# Selenium Deficiency-Induced Oxidative Stress Causes Myocardial Injury in Calves by Activating Inflammation, Apoptosis, and Necroptosis

**DOI:** 10.3390/antiox12020229

**Published:** 2023-01-19

**Authors:** Lei Lei, Jing Mu, Yingce Zheng, Yun Liu

**Affiliations:** 1Key Laboratory of Comparative Medicine, Department of Veterinary Surgery, College of Veterinary Medicine, Northeast Agricultural University, Harbin 150030, China; 2College of Life Science, Northeast Agricultural University, Harbin 150030, China

**Keywords:** Se deficiency, oxidative stress, inflammation, NF-κB, MAPK, apoptosis, necroptosis

## Abstract

Selenium (Se) is essential for human and animal health, but there have been few studies on the mechanisms of injury in dairy cows with Se deficiency. This study aimed to evaluate the effects of Se deficiency on myocardial injury in weaned calves. The Se-D group had significantly lower myocardial Se concentrations than the Se-C group. Histological analysis indicated that Se deficiency induced a large area of necrosis in the myocardium, accompanied by inflammatory changes. Se deficiency significantly decreased the expression of 10 of the 21 selenoprotein genes and increased the expression of SEPHS2. Furthermore, we found that oxidative stress occurred in the Se-D group by detection of redox-related indicators. Additionally, TUNEL staining showed that Se deficiency causes severe apoptosis in the myocardium, which was characterized by activating the exogenous apoptotic pathway and the mitochondrial apoptotic pathway. Se deficiency also induced necroptosis in the myocardium by upregulating MLKL, RIPK1, and RIPK3. Moreover, Se-deficient calves have severe inflammation in the myocardium. Se deficiency significantly reduced anti-inflammatory factor levels while increasing pro-inflammatory factor levels. We also found that the NF-κB pathway and MAPK pathway were activated in Se-deficient conditions. Our findings suggest that Se deficiency causes myocardial injury in weaned calves by regulating oxidative stress, inflammation, apoptosis, and necroptosis.

## 1. Introduction

Selenium (Se), an essential biological trace element, is mainly involved in selenoprotein synthesis in the form of selenocysteine. It has been demonstrated that tRNA^Sec^ deletion mice showed a lethal phenotype because of the lack of selenoprotein synthesis [1]. Most of the known selenoproteins are associated with redox events. In addition, the specific functions of selenoproteins include control of cytoskeleton assembly, formation and quality control of structural disulfide bonds, hormone activation and inactivation, selenoprotein synthesis, Se transport, and endoplasmic reticulum-related degradation [2]. Se deficiency causes a decrease in selenoprotein expression and activity, exacerbation of the redox byproduct toxicity, and finally damage to a variety of tissues.

There are benefits to human and animal health from adequate intake of Se or supplementation within safe limits [3]. However, up to 1/7 of the population worldwide has a low Se diet due to uneven Se distribution. As an important dairy feeding area in northeastern China, the Songneng Plain is also a Se deficiency region [4]. Even though Se supplementation has been emphasized in most farms, Se deficiency still occurs frequently. Affected adult cattle are usually treated by Se supplementation because the symptoms are less severe compared to those in calves [5]. However, in weaned calves, Se deficiency usually causes severe symptoms and even death due to food replacement and the relative fragility of the organism [6]. The prevalence of Se deficiency in agricultural areas will increase globally in the coming decades, due to changes in climate and soil organic carbon content [7].

Oxidative stress occurs through two main pathways: 1. low expression or decreased activity of antioxidant enzymes, resulting in a large accumulation of ROS produced by mitochondrial respiration [8,9]; 2. redox-related enzymes such as NOX2, COX-2, and lipoxygenases produce large amounts of ROS in specific metabolic processes [10]. It has been demonstrated that ROS is closely correlated with the NF-κB pathway. The NF-κB family of transcription factors not only plays a central role in inflammation and immunity but is also associated with cell growth, differentiation, development, and apoptosis. NF-κB signaling plays a dual role in oxidative stress [11]. On the one hand, NF-κB increases antioxidant enzyme expression to protect cells from death; on the other hand, more ROS induced by COX-2 and iNOS promote oxidative stress, ultimately leading to cell death [12]. Oxidative stress also activates the MAPK pathway [13], which regulates apoptosis, necroptosis, cell proliferation, and inflammatory gene expression. The role of MAPK in apoptosis is not uniform, it can be both anti- and pro-apoptotic depending on activation kinetics and stimulus type [14].

The Impaired cells were eliminate” thr’ugh regulated cell death (RCD) to shield the surrounding cells from damage. Apoptosis is a non-lytic, non-inflammatory death [15]. This process is dependent on the activity of caspase family proteins, including initiators caspase-8 and caspase-9, and executors caspase-3 and caspase-7. It is commonly believed that there are two major pathways of apoptosis: the intrinsic or mitochondrial pathway and the extrinsic or death receptor pathway, and these two pathways are intertwined. Necroptosis is another type of RCD that results in the loss of cytoplasmic membrane integrity and tissue inflammation [16]. Necroptosis is dependent on RIPK3 activity, and it is not only found in all vertebrates [17]. In the current study, we investigated whether this cell death modality is triggered in cows under Se-deficient conditions.

Se is essential for human and animal health, but there have been few studies on the mechanisms of injury in dairy cows with Se deficiency. In this study, Se concentrations, selenoprotein expression, histopathological changes, redox capacity, inflammation, apoptosis, and necroptosis were investigated to reveal the injury mechanism in the myocardium of Se-deficient weaned calves.

## 2. Materials and Methods

### 2.1. Animals and Experimental Design

All procedures in this study were approved by the Animal Care and Use Committee of the Institute of Animal Science; Northeast Agricultural university (SRM-11)

In September and October, a farm in Heilongjiang province had a large number of weaned calves with diarrhea, generalized edema, and death. The affected cows had no bacterial, viral, or parasitic infections according to the autopsies and the laboratory results, but their dietary Se levels were lower than normal. The Se content in the hearts of the affected cows was detected using inductively coupled plasma-mass spectrometry (ICP-MS), and the cows were found to be severely deficient in Se. The mildly symptomatic cattle in the herd were in remission after treatment with Se supplementation. Five affected female Holstein calves at 60 ± 3 d of age (67.22 ± 3.43 kg) from this farm were included in the Se-deficient group (Se-D group), which were confirmed to be Se-deficient and had no other diseases, while five healthy female Holstein cows at 59 ± 2 d of age (69.62 ± 1.22 kg) from other farms were included in the control group (Se-C group). All calves had free access to water and basal diet during their lifetime. Then, all the cows were slaughtered through electric shock and exsanguinated. The heart tissues were rapidly collected after being washed with ice-cold isotonic saline and stored at −20 °C for transport to the laboratory. All tissues were divided into two parts, one of which was trimmed to 1 × 1 × 1 cm square and fixed in 4% paraformaldehyde neutral buffer (pH 7.4–7.6) for histological examination, and the rest was minced, snap-frozen in liquid nitrogen, and then stored at −80 °C for storage.

### 2.2. Analysis of Se Concentrations in Tissues

Se concentrations in the myocardium were detected using the Agilent 7800 ICP-MS (Santa Clara, CA, USA), referring to the national standard (GB 5009.268) for the specific procedure.

### 2.3. TUNEL Staining and Histopathological Examination of the Myocardium

The formalin-fixed tissues were paraffin-embedded after proper dehydration and cut into 5 μm slices. Tissue damage was observed under light microscopy after hematoxylin and eosin staining. Paraffin-embedded tissues were stained with TUNEL using a commercial TUNEL kit purchased from Roche (Basel, Switzerland, cat. No. 11684817910). Five sections from each group were chosen at random and photographed using a fluorescence microscope in 200× field. The cells were counted using ImageJ to calculate the apoptotic index (Apoptotic index = number of TUNEL positive nuclei/total number of nuclei × 100%).

### 2.4. Redox Parameters and Inflammatory Cytokine Analysis

Minced myocardium (approximately 100 mg) was homogenized in nine-fold precooled normal saline using an Ultra-Turrax homogenizer at a low temperature. The homogenates were centrifuged at 3000 rpm for 15 min at 4 °C, and the resultant supernatants were collected for subsequent measurements. Redox-related parameters were measured using commercial kits from the Nanjing Jiancheng Bioengineering Institute of China. GSH concentration (A006-2-1), H_2_O_2_ concentration (A064-1-1), total antioxidant capacity (A015-1-1), and Thioredoxin Reductase activity (A119-1-1) were measured by spectrophotometric method; MDA concentration (A003-1-1) was measured by the thiobarbituric acid method; superoxide dismutase activity (A001-1-1) was measured by the hydroxylamine method; catalase activity (A007-1-1) was measured by the ammonium molybdate method. Inflammatory cytokines (including IL-1β (JM-00567B1), IL-6 (JM-00561B1), IL-8 (JM-08328B1), IL-10 (JM-00568B1), and IL-12 (JM-08322B1)) were measured with commercial enzyme-linked immunosorbent assay kits from Jiangsu Jingmei biotechnology Institute of China. The OD_450_ was detected using a microplate reader (BioTek Epoch, Winooski, VT, USA). The R^2^ of the standard curve was more than 0.9900, and the intra- and inter-plate coefficients of variation for cytokines were less than 15%. The detailed steps were performed according to the manufacturer’s instructions. All redox and inflammation-related parameters were normalized to protein concentrations, which were measured using the BCA method (Meilunbio, Dalian, China, cat. No. MA0082-2).

### 2.5. Real-Time PCR and Western Blot Analysis

RNA was extracted from heart tissue using the TRIZOL extraction method. To obtain purity and integrity of RNA, OD_260_/OD_280_ and OD_260_/_230_ of RNA were detected using Nano Drop (Nano Drop technologies, Wilmington, NC, USA), and 18S rRNA and 28S rRNA bands of RNA were observed using electrophoresis (1.2%). The cDNA was obtained using a PrimeScript RT reagent kit (Takara, Dalian, China, cat. No. RR047A) according to the manufacturer’s instructions. The primers for the selenoprotein, inflammation, apoptosis, and necroptosis related genes, as well as the housekeeping gene GAPDH, are shown in Table 1. All primers were designed using Oligo 7 (DBA Oligo, Inc., Cascade, CO, USA). The PCR products were run in an agarose gel to ensure the presence of a single specific amplification product and confirm specificity of RT-PCR. The melting curves had sharp, single peaks with the expected TM values. The cDNA templates were amplified using 2× SYBR Green qPCR Master Mix reagents (Bimake, Shanghai, China, cat. No. B21203) and gene expression was detected using the Roche LightCycler 480II system (Basel, Switzerland). A two-step method was used in the reaction process, including pre-denaturation at 95 °C for 5 min; amplification with 40 cycles, each cycle including 95 °C for 15 s and 60 °C for 40 s; and a melting reaction. Three replicate wells were set up for each sample to ensure the accuracy of the results. The 2^−ΔΔCt^ method was used for quantification analysis.

Minced myocardium (approximately 50 mg) was homogenized in 500 μL of RIPA lysis buffer (Beyotime bio, Shanghai, China, cat. No. P0013B)/PMSF (Beyotime bio, Shanghai, China, cat. No. ST506) mixture (100:1) using an Ultra-Turrax homogenizer at low temperature. The homogenates were centrifuged at 12,000 rpm for 10 min at 4 °C, and the resultant supernatants were collected. The total protein concentration was calculated with a BCA Protein Assay Kit, then diluted in SDS-PAGE sample loading buffer (Beyotime bio, Shanghai, China, cat. No. P0015L) and boiled for 15 min. The final concentration of the protein sample was 3 μg/μL. Equal amounts of protein (30 μg protein/line) were loaded and separated by 6–15% sodium dodecyl sulfate-polyacrylamide gel electrophoresis depending on the size of the target protein. After electrophoresis, the proteins were transferred to 0.2 μm nitrocellulose filter membrane (PALL, New York, NY, USA, cat. No. 66485), blocked in 5% (*w*/*v*) non-fat milk or 10% (*w*/*v*) BSA for 2 h and incubated overnight at 4 °C with the primary antibodies. Primary antibodies used in this experiment as follow: GPX1 (WL02497a), β-tubulin (WL01931), COX-2 (WL01750), iNOS (WL0992a), Bax (WL01637), Bak (WL0129a), Bcl-2 (WL01556), Caspase-3 (WL02117), Caspase-7 (WL02360), Caspase-8 (WL03426), Caspase-9 (WL03421), XIAP (WL03561), APAF1 (WL04536), TNF-α (WL01581), TNFR1 (WL01414), TRAF2 (WL02846), IKK α/β (WL01900), P-IκBα (WL02495), P-P50 (WL01866), P65 (WL01980), P-P65 (WL02169), P-ERK1/2 (WLP1512), P-JNK (WL01813), P-P38 (WLP1576), cIAP1 (WL03666), cIAP2 (WL01254), and c-FILP (WL02485) were purchased from Wanleibio (Shenyang, China); HIF-1α (BS-0737R), RIPK1 (BS-5805R), and RIPK3 (BS-3551R) were purchased from Bioss (Beijing, China); Txnrd3 (19517-1-AP) and MLKL (21066-1-AP) were purchased from Protintech (Rosemont, IL, USA); GAPDH (AC001) was purchased from ABclonal (Wuhan, China); Cleaved Caspase-3 (9661T) was purchased from Cell Signaling Technology (Beverly, MA, USA); GPX4 (ab41787) was purchased from Abcam (Cambridge, UK). β-tubulin or GAPDH was chosen as the internal reference according to the protein size, respectively. The membrane was washed in TBST and incubated with Goat Anti-Rabbit IgG (H + L) (Abclonal, Wuhan, China, cat. No. AS014) for 2 h at room temperature. Immuno-reactive bands were visualized with a ECL luminescence reagent (Meilunbio, Dalian, China, cat. No. MA0186-1). Protein bands were photographed using a Tanon-5200 gel imaging system. ImageJ (NIH, Bethesda, MD, USA) was used to analyze the blot signal and density.

### 2.6. Statistical Analysis

The mRNA (*n* = 5) and protein (*n* = 3) data were normalized using internal reference. The statistical differences of all the data in the experiment were assessed using SPSS 22.0 (SPSS Inc., Chicago, IL, USA). The data were tested for normality using the Shapiro–Wilk test. Normal distributed data were analyzed using two-tailed Student’s *t*-tests and skewed distributions were analyzed utilizing Mann–Whitney U tests, as required. Results are presented as mean ± standard deviation (SD). Significant differences were defined as * 0.01 < *p* < 0.05, ** 0.001 < *p* < 0.01, and *** *p* < 0.001. All figures were prepared with GraphPad Prism 5.0 (GraphPad Software Inc., La Jolla, CA, USA).

## 3. Results

### 3.1. Effect of Se Deficiency on Myocardial Se Concentration and Selenoprotein Expression in Weaned Calves

As shown in Figure 1A, the heart Se concentration in the Se-D group was 75.61% lower than in the Se-C group (*p* < 0.001). Figure 1B,C show the effect of Se deficiency on the relative expression of myocardial selenoprotein mRNA. Compared with the Se-C group, Se deficiency significantly altered 11 of the 21 selenoprotein genes in the myocardium of Se-D group. The relative mRNA levels of SEL-15, SEL O, and SEL S in the Se-D group were 55.0%, 42.7%, and 44.0%, respectively, of those in the Se-C group (0.01 < *p* < 0.05); the relative mRNA levels of SEL P, SEL T, and SEL W in the Se-D group were 55.4%, 55.3%, and 67.2%, respectively, of those in the Se-C group (0.001 < *p* < 0.01); the relative mRNA levels of GPX1, GPX4, SEL X, and DIO1 in the Se-D group were 34.9%, 18.9%, 21.0%, and 9.6%, respectively, of those in the Se-C group (*p* < 0.001). However, the transcript abundance of SEPHS2 in the Se-D group was 6.36 times greater than that in the Se-C group (*p* < 0.001). In addition, there were no significant differences in the relative mRNA levels of GPX3, SEL H, SEL I, SEL K, SEL M, SEL N, TXNRD1, TXNRD2, TXNRD3, and DIO2. Western blotting (Figure 1D,E) indicated that the protein expression of GPX1 and GPX4 in the Se-D group was 63.3% and 66.0%, respectively, of those in the Se-C group (*p* < 0.05); TXNRD3 in the Se-D group was 76.9% of that in the Se-C group, but the difference was not significant.

### 3.2. Histopathological Changes in the Myocardium of Weaned Calves under Se-Deficient Conditions

Myocardial histopathology in the Se-C group showed well-aligned myocardial fibers with intact cell structure. The myocardium of the Se-D group developed a significant region of necrosis, with myocardial fiber fracture, lysis, and loss of texture, as well as nuclei lysis and disappearance, all of which were accompanied by inflammatory cell infiltration (Figure 2).

### 3.3. Effect of Se Deficiency on Myocardial Oxidative Stress Levels in Weaned Calves

The results of myocardial oxidative stress-related indicators are shown in Figure 3A. The content of GSH and T-AOC in the Se-D group was 67.6% and 82.2% of that in the Se-C group respectively (0.01 < *p* < 0.05); the activity of SOD in the Se-D group was 85.0% of that in the Se-C group (0.001 < *p* < 0.01); the activity of CAT and TrxR in the Se-D group was 34.2% and 49.9% of that in the Se-C group (*p* < 0.001); the content of MDA and H_2_O_2_ in the Se-D group was 1.49 and 1.79 times higher than that in the Se-C group (0.001 < *p* < 0.01).

RT-PCR results (Figure 3B) showed that COX-2, HIF-1α, and iNOS mRNA expression were elevated in the Se-D group, 22.37, 24.37, and 19.79 times higher than those in the Se-C group, respectively (*p* < 0.001); WB results (Figure 3C,D) were in accordance with RT-PCR results. The protein expression of COX-2 and HIF-1α were elevated in the Se-D group, 1.69 and 2.01 times higher than those in the Se-C group (0.01 < *p* < 0.05); iNOS in the Se-D group was 1.53 times higher than that in the Se-C group (0.001 < *p* < 0.01).

### 3.4. The Apoptotic Pathways Were Activated in the Myocardium of Se-Deficient Calves

TUNEL staining was performed on paraffin sections to investigate the effect of Se deficiency on apoptosis in calves’ myocardium. As shown in Figure 4A,B, there were no apoptotic cells in the Se-C group, and a significant amount of apoptosis occurred in the Se-D group, with an apoptotic index of 36.95–68.40%. To further verify the results, we examined the expression levels of seven apoptosis-related mRNAs. As shown in Figure 4C, Caspase-3, Caspase-7, Caspase-8, Caspase-9, Bax, and Bak were 17.25, 6.30, 7.53, 13.65, 33.51, and 14.73 times higher in the Se-D group compared with the Se-C group, respectively (*p* < 0.001); Bcl-2 was significantly decreased, the Se-D group was 49.09% of the Se-C group (*p* < 0.001). In addition, we also examined ten apoptosis-related proteins by WB. The results are shown in Figure 4D and 4E, Bax, Bak, cleaved Caspase-3, Caspase-7, Caspase-8, Caspase-9, and APAF1 were significantly increased in the Se-D group (0.01 < *p* < 0.05), which were 2.00, 1.51, 1.62, 1.37, 1.56, 2.08, and 1.49 times higher compared with the Se-C group, respectively; Caspase-3 in the Se-D group was 1.53 times higher than that of the Se-C group (*p* < 0.001). However, XIAP in the Se-D group was 71.77% of that in the Se-C group (0.01 < *p* < 0.05), and Bcl-2 in the Se-D group was 49.12% of that in the Se-C group (0.001 < *p* < 0.01). This result was in accordance with the RT-PCR results, indicating that severe apoptosis occurred in the Se-D group.

### 3.5. The Activated NF-κB and MAPK Pathways Induced Inflammation in the Myocardium of Se-Deficient Calves

To investigate the effect of Se deficiency on inflammation-related indicators in the myocardium of calves, we examined five inflammatory factors by ELISA. As shown in Figure 5A, pro-inflammatory cytokines IL-8 and IL-12 were 1.36 and 1.31 times higher in the Se-D group than in the Se-C group (0.01 < *p* < 0.05); IL-1β and IL-6 in the Se-D group were 1.30 and 1.28 times higher in the Se-D group than in the Se-C group (0.001 < *p* < 0.01); while anti-inflammatory cytokine IL-10 in the Se-D group was 78.73% of that in the Se-C group (0.01 < *p* < 0.05). 

The effect of Se deficiency on the relative expression of inflammatory factors mRNA was in accordance with the ELISA results (Figure 5B). Pro-inflammatory factor mRNA levels of IL-1β, IL-6, IL-8, and IL-12 in the Se-D group were 16.42, 118.85, 19.57, and 11.64 times, respectively, higher than those in the Se-C group (*p* < 0.001); anti-inflammatory factors relative mRNA levels of IL-10 and TGF-β1 in the Se-D group were 27.07% and 37.03% of those in the Se-C group (*p* < 0.05). The NF-κB pathway-related genes TNF-α, TNFR1, P50, and P65 were 15.21-, 5.88-, 19.75-, and 9.24-fold upregulated in the Se-D group compared to the Se-C group (*p* < 0.001). The results of WB were consistent with the RT-PCR results (Figure 5C,D). The protein levels of TNFR1, P65, and P-P65 in the Se-D group were 2.30, 2.50, and 1.82 times higher than in the Se-C group. TNF-α, TRAF2, IKKα/β, P-IκBα, and P-P50 were upregulated in the Se-D group, but the differences were not significant. The MAPK pathway-related genes ERK2, JNK, and P38 were 8.83, 11.38, and 6.09 times higher in the Se-D group compared to those in the Se-C group, respectively (*p* < 0.001). The results of WB were in accordance with the RT-PCR results (Figure 5E,F). The protein levels of P-ERK1/2 and P-JNK in the Se-D group were 2.90 and 1.44 times higher than those in the Se-C group (*p* < 0.01), and P-P38 was elevated, but the difference was not significant.

### 3.6. Necroptosis Was Activated in the Myocardium of Se-Deficient Calves

To investigate the effect of Se deficiency on necroptosis in the myocardium of calves, we examined the mRNA expression levels of four necroptosis-related genes, and the results are shown in Figure 6A. The mRNA level of cFLIP in the Se-D group was 2.99 times higher than in the Se-C group (0.01 < *p* < 0.05); MLKL, RIPK1, and RIPK3 were 15.34, 10.18, and 9.99 times higher than in the Se-C group, respectively (*p* < 0.001). In addition, we also examined six proteins related to necroptosis by WB, and the results are shown in Figure 6B,C. The protein level of RIPK3 in the Se-D group was 1.80 times higher than in the Se-C group (0.01 < *p* < 0.05); cIAP2, cFLIP, RIPK1, and MLKL were 1.23, 3.01, 2.44, and 1.64 times higher than in the Se-C group, respectively (0.001 < *p* < 0.01), associated with a slight but not significant increase in cIAP1. The results of WB were in accordance with the RT-PCR results, indicating that necroptosis occurred in the Se-D group.

## 4. Discussion

The heart is the major organ of injury in Se deficiency. In our experiment, extensive myocardial necrosis was observed in the Se-D group, which is consistent with the white muscle disease described in the literature [18]. One of the main factors contributing to calf death may be heart failure brought on by such a significant cardiac injury. The damage caused by Se deficiency is irreversible because cardiomyocytes are almost irreducible in vivo. It is important to explore the mechanism of myocardial injury for dairy calf health.

Se is an important component of the active center in selenoproteins. The decrease of Se content in tissues impairs selenoprotein biological functions by downregulating expression and activity [19]. The results of this experiment showed that the Se concentration, the expression of GPX1 and GPX4, and the activity of the Se-containing enzymes TrxR were significantly reduced in the Se-D group. GPX1 is the most abundant selenoprotein in mammals and is essential for intracellular hydrogen peroxide homeostasis [20]. It has been shown that knockout of the GPX1 gene causes sensitivity to oxidative stress in mice [21]. GPX4 is involved in the reduction of membrane-associated phospholipid hydroperoxides and inhibits lipid peroxidation [22]. Thioredoxin reductases (TrxRs), together with thioredoxin, constitute the intracellular disulfide bond reduction system [23]. Although there was no change in the gene and protein levels, the activity of TrxR was significantly reduced in this experiment, which may be due to a Se deficiency resulting in the misincorporation of cysteine into protein synthesis, leading to reduced antioxidant capacity [24,25]. Se deficiency also affects the expression of some selenoprotein in mRNA levels. In this experiment, GAPDH was chosen as a housekeeping gene to investigate the transcript levels of 21 selenoprotein genes in the hearts [26]. The results showed that the expression of mRNAs of 10 selenoprotein genes was significantly reduced, including SEL-15, SEL O, SEL S, SEL P, SEL T, SEL W, GPX1, GPX4, SEL X, and DIO1. The expression of GPX1 and GPX4 proteins was in accordance with their mRNA. All the above results were basically consistent with previous studies [27,28]. The variation in selenoprotein mRNA differences may be related to the fact that selenoproteins can be regulated at the level of transcriptional, translational, and post-translational modifications [29]. Depending on the Se consumed, selenoprotein mRNAs exhibit different species and tissue specificities [30]. In Se-deficient conditions, selenoproteins are synthesized according to a strict hierarchy depending on selenoprotein mRNA stability [31], the interaction of the 3’UTRs of SECIS with the coding sequence [32], and the affinity of SBP2 for different selenoproteins [33]. Previous studies have shown that SEL-15 [34], SEL S [35], SEL P [36,37], SEL W, SEL T [38], and SEL X [39] are all associated with the antioxidant capacity of the organism. It has been shown that the downregulation of SEL O may be an important cause of endemic osteoarthrosis due to Se deficiency [40]. The deiodinase enzyme 1 (DIO1) plays an important role in circulating thyroid hormone levels and activity, which may also contribute to skeletal muscle lesions in Se deficiency [41]. Selenium phosphate synthase 2 (SEPHS2) plays a self-regulatory role in selenoprotein synthesis by catalyzing the synthesis of active Se donor-selenium phosphate [42]. In this experiment, SEPHS2 expression was significantly increased, most likely in response to tissue demand for other selenoproteins and thus compensating for the elevation [43]. We failed to detect GPX2, GPX6, SEL V, and DIO3 in this study because these four selenoprotein genes were minimally or not expressed in the heart, which is also consistent with the literature [26,44]. In summary, most of the low-expressed selenoprotein genes detected in this experiment are related to redox, suggesting that we need to carry out further validation of this part.

The activities of antioxidant-related enzymes (CAT, SOD, and TrxR), the content of GSH, and the total antioxidant capacity (T-AOC) were significantly decreased while the content of H_2_O_2_ was significantly increased in the Se-D group, suggesting that the hearts of Se-deficient calves were in a state of oxidative stress. H_2_O_2_ is a signaling molecule for cell proliferation, apoptosis, the stress response, and mitochondrial functions [45]. Lipid peroxidation also occurs in oxidative stress, producing a variety of highly reactive aldehydes, including malondialdehyde (MDA), which has a long metabolic cycle and acts as a second toxic messenger in the initial free radical event. The large accumulation of these aldehydes can be damaging to the organism [46]. The significantly increased MDA in the Se-D group may be associated with GPX4 downregulation. The deletion of GPX4 leads to lipid peroxidation due to its function in phospholipid peroxides decomposition [22]. The large amount of ROS and other byproducts produced in oxidative stress causes direct damage to lipids, proteins, and DNA, which may be one of the underlying causes of Se deficiency disease.

In the study, iNOS was found to be significantly elevated in the Se-D group at both gene and protein levels. Inducible nitric oxide synthase (iNOS) is responsible for NO synthesis. An abnormal increase in NO levels or a decrease in SOD activity leads to the formation of large amounts of peroxynitrite by combining NO with superoxide. Peroxynitrite is a strong oxidizing agent that damages a variety of organs [47]. Peroxynitrite also activates the ERK pathway to exert pro-apoptotic effects [13]. Increased endogenous NO promotes the synthesis of TNF-α and IL-1β through the MAPK-ERK pathway, inducing transcription and protein synthesis of HIF-1α [48]. NO also accumulates HIF-1α and improves its stability and activity via post-translational modification [49]. Significant elevation of HIF-1α in gene and protein levels was also observed in the Se-D group in this study. HIF-1α plays a key role in the adaptive regulation of energy metabolism in mammalian [50]. Although HIF-1α resists hypoxia-related injury, prolonged HIF-1α signaling leads to increased inflammation, oxidative stress, and fibrosis [51,52,53]. It is shown that ROS regulates HIF-1α transcription and translation by inducing miR-21 activation of the PI3K/AKT and ERK pathways [54], which are linked to metabolic disorders and inflammation. In this study, we found that the ERK, JNK, and P38 pathways were activated in Se-deficiency. Stimulation of ROS or reduction of TrxR activity induces dissociation of oxidized TRX1 from the “ASK1 signalosome”. Activated ASK1 is involved in regulating inflammation and cell death via MAPK-JNK [55]. H_2_O_2_ also rapidly activates the MAPK pathway to induce apoptosis [13]. COX-2 is a key enzyme mediator of oxidative stress and was found significantly elevated in the Se-D group at both mRNA and protein levels in this study. Lipid peroxide and peroxynitrite can catalyze COX activity, while COX-2 also exacerbates oxidative stress by generating superoxide in turn [56]. As a downstream gene of NF-κB and MAPK, COX-2 is abundantly expressed in inflammatory tissues [57]. The p38 pathway is critical for cytokine-induced COX-2 mRNA stability. The inhibition of the p38 pathway may be responsible for the anti-inflammatory effects of dexamethasone [58].

Inflammation is usually triggered when the body is under oxidative stress. The expression of cardiac pro-inflammatory factors IL-1β, IL-6, IL-8, IL-12, and TNF-α was significantly increased in this study, while anti-inflammatory cytokines IL-10 and TGF-β were significantly decreased, indicating a severe inflammatory response in Se-deficient calves’ hearts. Next, we found the NF-κB pathway was activated in Se deficiency conditions. Previous research has shown that Se deficiency promotes oxidative stress-induced mastitis via activating the NF-κB pathways in dairy cow [59]. The NF-κB pathway is not active in normal cells, and the P50/P65 dimer is present in the cytoplasm in combination with the inhibitor IκB. TNF-α activated the TRAFs complex via its receptor TNFR1 to phosphorylate downstream IKKs, causing IκBs to ubiquitinate and degrade by phosphorylating and releasing P50/P65 complexes. P50/P65 complexes rapidly enter the nucleus to regulate transcription and enhance the expression of inflammatory cytokines, chemokines, growth factors, cell adhesion molecules, or acute phase proteins [11]. Oxidative stress triggers inflammation, and inflammatory mediators participate in the oxidative stress, contributing to a vicious cycle.

In our previous study, we found that Se deficiency induces apoptosis in calf liver [60]. Normal cardiomyocytes are in a state of terminal differentiation and undergo little apoptosis [61]. TUNEL staining, however, revealed that severe apoptosis occurred in the Se-D group in this study. Our study shows that apoptosis under Se-deficiency occurs mainly through two pathways: 1. the exogenous apoptotic pathway can be activated by TNFR1, leading to Caspase-8 homodimerization, activation, autoproteolytic processing, and release from the complex. Fully activated Caspase-8 cleaves the executioner Caspases-3 and Caspases-7 to induce apoptosis; 2. the mitochondrial apoptotic pathway is mediated by the BCL-2 effector proteins Bax and Bak to induce mitochondrial outer membrane permeabilization (MOMP). Activation of BCL-2 effector proteins is regulated by the balance between proapoptotic and antiapoptotic Bcl-2 proteins [62]. MOMP releases Cyt C and IAPi into the cytosol, where Cyt C binds and activates APAF-1 to recruit the initiator Caspase-9, forming the apoptosome and activates Caspases-3 and Caspases-7 to mediate apoptosis. IAPi block the suppressive effects of the XIAP, allowing the caspases to function [63].

Necroptosis is an independent mechanism of cell death and should not be considered an alternate mechanism that is engaged only when apoptosis fails. Individual cells appear to undergo only one type of cell death, apoptosis or necroptosis, and it is unclear how the cell decides which type of cell death it will engage in [64]. This study revealed that Se deficiency induces necroptosis in cardiomyocytes. The pathway does not depend on caspases but on the activity of RIPK3, the only known activator of MLKL [65]. When TNF-α signaling is activated, ligation of TNFR1 induces the recruitment of TRADD, TRAF2, cIAP1/2, and RIPK1. After being deubiquitylated by cIAP1/2, RIPK1 has a dual role in necroptosis [66]: on the one hand, it leaves the complex and associates with RIPK3 through homotypic RHIM domain interactions, resulting in RIPK3 activation; on the other hand, RIPK1 mediates the expression of proinflammatory and prosurvival genes, including cFLIP, meanwhile, it recruits the FADD–proaspase8–cFLIP complex to RIPK3 to inhibit necroptosis. Autoactivated RIPK3 recruits and activates MLKL, which induces plasma membrane pore formation and necroptosis. FLIP has similar structural features to caspase-8 but lacks caspase-8 activity. The function of cFLIP correlates with its expression level [63]: at low levels, it promotes caspase-8 dimer formation and apoptosis; at high levels, it forms heterodimers with caspase-8 to inhibit both apoptosis and necroptosis. Interestingly, the significantly upregulated levels of cFLIP seem not to have effectively inhibited apoptosis and necroptosis in this study. The upregulation of cFLIP may reflect the protective effect of the NF-κB pathway, which exerts an anti-apoptotic function to reduce cell death [67]. The specific mechanism needs to be explored in the future.

In summary, the mechanism of myocardial injury due to Se deficiency is very complex. In this experiment, we found that Se deficiency caused a decrease in selenoprotein expression and antioxidant capacity, as well as inflammation, apoptosis, and necroptosis in weaned calves under feeding conditions. Our study provided a reference for further research on the injury models of Se deficiency disease in dairy cows and a theoretical basis for mitigating the damage caused by Se deficiency in clinical practice. The current results were limited to the mechanisms of injury mentioned in this article and were unable to fully generalize to all pathological processes that occur in Se deficiency. Future research should look into whether Se deficiency causes myocardial injury in calves through other mechanisms, such as autophagy, ferroptosis, and so on.

## Figures and Tables

**Figure 1 antioxidants-12-00229-f001:**
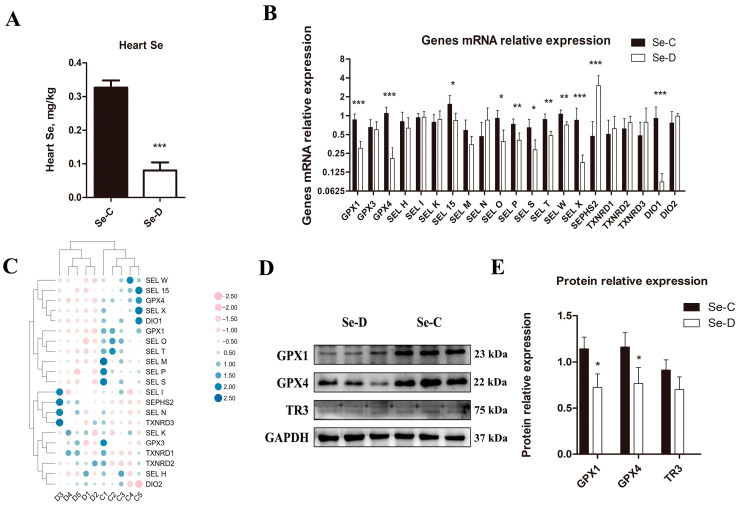
Se concentrations (**C**), relative mRNA (**A**,**B**) and protein (**D**,**E**) levels of selenoproteins in the Se-C group and Se-D group myocardium. Values are means ± SD. * 0.01 < *p* < 0.05, ** 0.001 < *p* < 0.01, and *** *p* < 0.001 different from Se-C, respectively.

**Figure 2 antioxidants-12-00229-f002:**
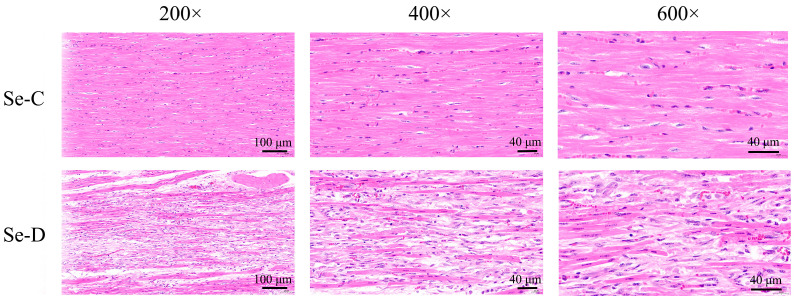
Hematoxylin and eosin staining in the Se-C group and Se-D group myocardium. Photomicrographs are shown at 200×, 400×, and 600× magnification.

**Figure 3 antioxidants-12-00229-f003:**
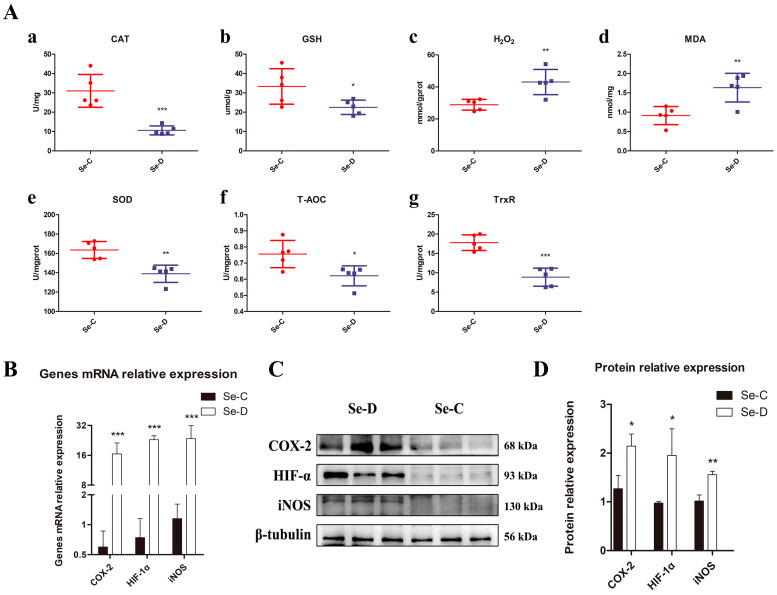
The activity of CAT, SOD, TrxR, the total antioxidant capacity, and the content of GSH, H_2_O_2_, MDA (**A**). Calves in the Se-C group are represented by red circles, while calves in the Se-D group are represented by blue squares. The relative mRNA (**B**) and protein levels (**C**,**D**) of COX-2, HIF-1α, and iNOS in the Se-C group and Se-D group hearts. Values are means ± SD. * 0.01 < *p* < 0.05, ** 0.001 < *p* < 0.01, and *** *p* < 0.001 different from Se-C, respectively.

**Figure 4 antioxidants-12-00229-f004:**
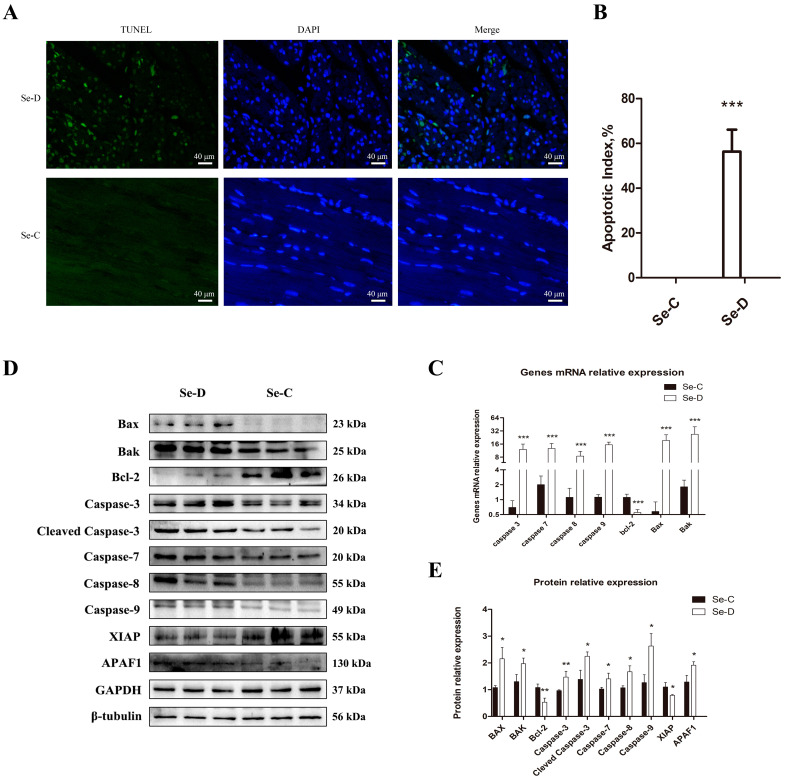
TUNEL staining (**A**,**B**) of myocardium and relative mRNA (**D**) and protein levels (**C**,**E**) of apoptosis in the Se-C group and Se-D group. Photomicrographs are shown at 400× magnification. Values are means ± SD. * 0.01 < *p* < 0.05, ** 0.001 < *p* < 0.01, and *** *p* < 0.001 different from Se-C, respectively.

**Figure 5 antioxidants-12-00229-f005:**
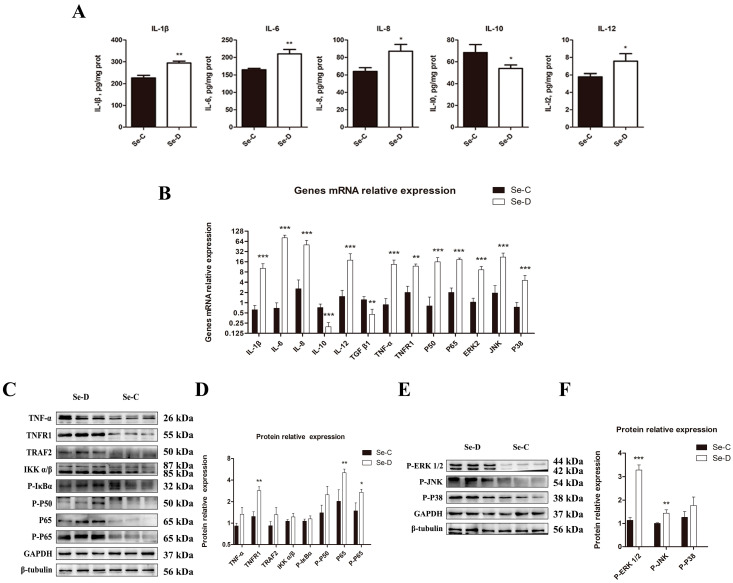
The content of the pro-inflammatory cytokines IL-1β, IL-6, IL-8, IL-12, and the anti-inflammatory cytokines IL-10 (**A**). The relative mRNA (**B**) and protein levels (**C**–**F**) of inflammation in the Se-C group and Se-D group myocardium. Values are means ± SD. * 0.01 < *p* < 0.05, ** 0.001 < *p* < 0.01, and *** *p* < 0.001 different from Se-C, respectively.

**Figure 6 antioxidants-12-00229-f006:**
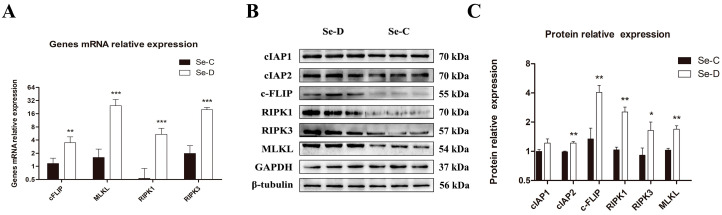
Relative mRNA (**A**) and protein levels (**B**,**C**) of necroptosis in the Se-C group and Se-D group myocardium. Values are means ± SD. * 0.01 < *p* < 0.05, ** 0.001 < *p* < 0.01, and *** *p* < 0.001 different from Se-C, respectively.

**Table 1 antioxidants-12-00229-t001:** The primers used in the present study.

Gene	Serial Number	Forward Primer (5′–3′)	Reverse Primer (5′–3′)
DIO1	NM_001122593.2	85F: CTCCTGACGCTGTTTCCCAG	149R: GAAGTGTGGGTTCCTGGTCA
DIO2	NM_001010992.7	558F: GAACCAGGAAGACCGATGCG	670R: ACACGTTCAAAGGCTACCCC
DIO3	NM_001010993.3	107F: AGACTTCCTGTGCATCCGTA	363R: AGTCGAGGATGTGCTGGTTC
TXNRD1	NM_174625.5	20F: TCCCGGAGCCCTATGACTAC	219R: GGGCTTGACCTAACAACGCT
TXNRD2	NM_174626.3	357F: TCAGAACCACGTGAAGTCCC	560R: GATCCCATATTCCAAGGCAC
TXNRD3	NM_001192109.2	1289F: AAGGGACAGAGACAATCGAG	1377R: CACTGACGTTCACACCAATC
GPX1	NM_174076.3	89F: AGCCCTTCAACCTGTCCTCC	302R: CGTACTTCAGGCAATTCAGG
GPX2	NM_001163139.2	137F: ACTTCACCCAACTCAACGAG	340R: TTCAGGTAGGCAAAGACAGG
GPX3	BC149266.1	252F: ACTGCAGGAAGAGCTTGAAC	460R: GAGGTAGGAGGACAGGAGTT
GPX4	NM_174770.4	13F: CGTCTGTACCGCCTGCTCAA	212R: GTCAGTCTTGCCTCATTGCG
GPX6	NM_001163142.2	246F: GAATGCACTACAGGAGGAGC	496R: TCCCAGAAGAGCTGACTTGA
SEL H	NM_001164092.1	192F: CGAGCTGACGGTGAAGGTGA	344R: GTACTTCTTCAGCTCCTCCA
SEL I	NM_001075257.2	408F: TGTTGTCACCGTGTATTCCA	619R: AGGAAAGGTTCATACCAGGC
SEL K	NM_001037489.3	81F: CTGGGGAATAGCTGAGTTTG	251R: GGCCATTGGAGGAGGATTAG
SEL M	NM_001163171.3	175F: CAGGACATCCCACTCTACCA	255R: ATTCGCTCCAGTTCCTCAAAG
SEL N	NM_001114976.3	251F: ACATGTACATCAGCCCCGAG	451R: CAGTTGCGGAGGCCAGACAG
SEL O	NM_001163193.3	762F: CACGTTCCTCAGGTTTGGAT	862R: CTGATGACGTAGTCGAGCAT
SEL P	NM_174459.3	65F: AGGGCCAAAGCTCCTATTGT	271R: CGAGAAGAGATTCCTTGATG
SEL S	NM_001046114.3	145F: GACAGAAGATCGAAATGTGG	307R: TCCACCACCTTCACCAGACA
SEL T	NM_001103103.2	423F: CCAGTGTATGTCAACAGGTG	538R: TGCACGTTGAGCTTCATTTC
SEL V	NM_001163244.2	842F: TAGCTTCCATCAGTGGGAAC	1071R: GTAGCTTGACCTCATCCACA
SEL W	NM_001163225.1	22F: GTTTATTGTGGCGCTTGAGG	216R: CGGCCACCAGCTTCAGAAAC
SEL X	BC105188.1	7F: TTCTGCAGCTTCTTCGGAGG	168R: GATTGTGCTCTGGCCGCTTG
SEL 15	BT021503.1	271F: TGTGGATGAAAATTGGGGAG	394R: GCAATGTTCCCACTGTCGTC
SEPHS2	NM_001114732.3	965F: GCTTTGGCATTCTGGGTCAC	1203R: CGACGATCCAGGCTTGATGA
Bcl-2	NM_001166486.1	408F: CATCGTGGCCTTCTTTGAGT	592R: CAGGAGAAATCAAACAGGGG
Caspase 3	NM_001077840.1	357F: AAGCCATGGTGAAGAAGGAA	529R: GCCATGTCATCCTCAGCACC
Caspase 7	XM_002698509.5	611F: ACACTGATGCTAATCCCCGT	748R: AGGCTCTTCCCATGCTCATT
Caspase 8	NM_001045970.2	1283F: TCCAGTCACTTTGCCAGAAT	1411R: CGCAGTGTGAAAGTAGGTTG
Caspase 9	NM_001205504.2	267F: GAAGACCAGCAGACAAGCAG	418R: TCAGTGAATCCTCCAGAACC
Bak	NM_001077918.1	389F: CTCTGCTGGGCTTTGGCTAC	592R: ACAAACTGGCCCAACAAAAC
Bax	NM_173894.1	72F: CCTTTTGCTTCAGGGTTTCAT	265R: ACTCGGAAAAAGACCTCTCG
IL-1β	NM_174093.1	546F: GTCTTGTGTGAAAAAAGGTG	715R: ACGGGCCTTTCTTCGATTTG
IL-6	NM_173923.2	381F: CTACCTCCAGAACGAGTATG	481R: GTGGCTGGAGTGGTTATTAG
IL-8	BC103310.1	114F: AACACATTCCACACCTTTCC	261R: TCACAAATACCTGCACAACC
IL-10	NM_174088.1	251F: CGGAAATGATCCAGTTTTACC	457R: CTCATGGCTTTGTAGACACC
IL-12	NM_174356.1	702F: CAAACCAGACCCACCCAAGA	866R: GGCTGAGGTTTGGTCCATGA
TNF-α	NM_173966.3	75F: GGGCTCCAGAAGTTGCTTGT	175R: TGGGGACTGCTCTTCCCTCTG
TNFR1	NM_174674.2	266F: ACACTGCCTTGGAGAACCAT	402R: AGCCAGTTTCACCCCAGTAT
TGF β1	NM_001166068.1	327F: CCGCGTGCTAATGGTGGAAT	444R: CGTCTGCCCGAGAGAGCAAC
P50	NM_001076409.1	720F: TGACAGCAAAGCCCCCAATG	985R: CTCCGAAGCTGGACGAACAC
P65	NM_001080242.2	402F: CCAGACCAACAACAACCCCT	452R: GACGGCATTCAGGTCGTAGT
iNOS	DQ676956.1	117F: GACACAGGATGACCCCAAAC	316R: GACTTGCAAGAGAGATCCCC
HIF-1α	NM_174339.3	17F: GCGCGAACGACAAGAAAAAG	214R: AAATCACCAGCATCCAGAAG
COX-2	AF004944.1	96F: AGCTCTTCCTCCTGTGCCTG	283R: GCTGGTCCTCGTTCAAAATC
ERK2	NM_175793.2	182F: ACCAGACGTACTGCCAGAGA	440R: GGAAGGTTTGAGGTCACGGT
JNK	NM_001192974.2	8F: GAAGCAAGCGTGACAGCAAT	96R: TTCCTTGGGCTCCTGAACCT
P38	NM_001102174.1	505F: TTCGGACTGGCTCGACATAC	721R: TAATGAGATAAGCAGGGGGAGT
cFLIP	NM_001012281.1	220F: ATGGATACCCCGACAGTGGA	360R: TGGCTATCTTGCTTCGACCC
RIPK1	NM_001035012.1	1359F: CCAACCCCAGTCACCATACT	1595R: ACTGTCGCCAATCTGAATGC
RIPK3	NM_001101884.2	594F: CTCCAGGGCCAGTGATGTCTA	690R: AATGATGTCTGGGGCACTGAGG
MLKL	XM_024978879.1	488F: TGGAGGAAACCATCGAAGCC	762R: GCAGGATGTTGGGGGAATCA
GAPDH	NM_001034034.2	8F: AGGTCGGAGTGAACGGATT	194R: GGCCTTTCCATTGATGACGA

## Data Availability

The original contributions presented in the study are included in the article. Further inquiries can be directed to the corresponding authors.

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
