# Peer review of "Selenium Deficiency-Induced Oxidative Stress Causes Myocardial Injury in Calves by Activating Inflammation, Apoptosis, and Necroptosis"

_antioxidants, 2023, doi:10.3390/antiox12020229_

Round 1

Reviewer 1 Report

This work performed a comprehensive survey on selenium deficiency effects in calves. The experiments were performed using standard methods. The manuscript is written clearly and precisely. As a professional of the Se-insertion mechanism, I think this manuscript can be published after minor revision.

I was surprised by the very long discussion section. Some of the arguments could be omitted or supported by citing very recent papers of the authors and others. For example, I found two papers.

Biol Trace Elem Res 2022 Nov;200(11):4678-4689. doi: 10.1007/s12011-021-03059-5.

Selenium Deficiency Induces Apoptosis, Mitochondrial Dynamic Imbalance, and Inflammatory Responses in Calf Liver

Biol Trace Elem Res 2022 Jun;200(6):2716-2726. doi: 10.1007/s12011-021-02882-0.

Selenium Deficiency Promotes Oxidative Stress-Induced Mastitis via Activating the NF-κB and MAPK Pathways in Dairy Cow

I found a few typos in the text and pictures.

In line 67: processe -> processes ?

In line 280: Se -D -> Se-D ?

Do you need a period for every citation to Figures? e.g. Figure.2.

In Fig. 1D: The G of GAPDH is behind a picture?

In Fig. 3D: INOS -? iNOS ?

Will all pictures be replaced with high resolution ones?

Author Response

Dear reviewer,

Thank you very much for your careful review and valuable suggestions. The following changes have been made to the manuscript. We highlighted in blue all the changes in the main text.

Comment 1:

I was surprised by the very long discussion section. Some of the arguments could be omitted or supported by citing very recent papers of the authors and others. For example, I found two papers.

Biol Trace Elem Res 2022 Nov;200(11):4678-4689. doi: 10.1007/s12011-021-03059-5.

Selenium Deficiency Induces Apoptosis, Mitochondrial Dynamic Imbalance, and Inflammatory Responses in Calf Liver

Biol Trace Elem Res 2022 Jun;200(6):2716-2726. doi: 10.1007/s12011-021-02882-0.

Selenium Deficiency Promotes Oxidative Stress-Induced Mastitis via Activating the NF-κB and MAPK Pathways in Dairy Cow

Response 1:

The discussion section has been shortened. Some arguments have been omitted, some references have been updated, and your recommended papers have been cited in lines 400 and 410.

Comment 2:

I found a few typos in the text and pictures.

In line 67: processe -> processes ?

In line 280: Se -D -> Se-D ?

Do you need a period for every citation to Figures? e.g. Figure.2.

In Fig. 1D: The G of GAPDH is behind a picture?

In Fig. 3D: INOS -? iNOS ?

Response 2:

All typos mentioned in the comment 2 have been corrected in the revised version.

In line 51: processe -> processesï¼›

In line 263: Se -D -> Se-Dï¼›

We have revised all descriptions in the text when citing Figures. e.g. Figure.2 -> Figure 2ï¼›

We have modified Fig. 1D so that the G of GAPDH can be displayed correctly.

In Fig. 3D: INOS -> iNOS.

Comment 3:

Will all pictures be replaced with high resolution ones?

Response 3:

We have rechecked that the pictures in the re-uploaded manuscript are high-resolution. The first submission, in fact, is also in high-resolution. Please let us know if the pictures are still unclear this time.

Reviewer 2 Report

The manuscript is well-written, and the experiments are correctly performed. However, the authors should do other experiments to demonstrate apoptosis (different molecules) histologically in the myocardial samples. The quality of the histological images is low. Therefore, in my opinion, the authors should also improve the quality of the pictures they presented.  

Author Response

Dear reviewer,

Thank you very much for your careful review and valuable suggestions. Our manuscript has been checked by a native English-speaking and corrected for language and style. We highlighted in blue all the changes in the main text.

Comment 1:

However, the authors should do other experiments to demonstrate apoptosis (different molecules) histologically in the myocardial samples.

Response 1:

Additional experiments are difficult due to the impact of COVID-19, and we think that TUNEL staining, changes in mRNA, and WB assays can convincingly demonstrate apoptosis in myocardial samples. Please let us know in great detail if you consider we should include more experiments. 

Comment 2:

The quality of the histological images is low. Therefore, in my opinion, the authors should also improve the quality of the pictures they presented.  

Response 2:

We have rechecked that the pictures in the re-uploaded manuscript are high-resolution. The first submission, in fact, is also in high-resolution. Please let us know if the pictures are still unclear this time.

Reviewer 3 Report

This study aimed to evaluate the effects of Selenium (Se) deficiency on myocardial injury in weaned calves. Based on their results, the authors concluded that Se deficiency induced a large area of necrosis in the myocardium, accompanied by inflammatory changes.

The abstract section should be improved. The authors should follow the manuscript's structure: Background and aims, Material and methods, results, discussion, and conclusion. Moreover, it should be a total of about 200 words maximum. Please, change it accordingly, following the authors' guidelines (https://www.mdpi.com/journal/antioxidants/instructions).

The introduction section is too long: please, summarize the main background information in order to introduce the theme, describing clearly the aims. Please, revisit it.

Despite the material and methods section describe the main tools utilized in this study, it should be improved. I suggest inserting a schematic picture in order to clarify the experimental model. Moreover, the authors missed inserting several important pieces of information (the composition of each group, the middle age, the health status, the diet conditions, ....). Please, improve this section.

The results section should be improved. In Figure 2, the metric reference should be inserted (bar line). The statistical analysis should be completely described in supplementary files. Please, insert complete information in order to better analyze the discussed results.

Finally, the original images (inserted as a supplementary file) of the blots are strange: please, insert the gel images at the moment of the acquisition, without increasing the contrast of the bands.

The discussion section should be improved: the authors should insert several considerations about the limitations of the present study.

Author Response

Dear reviewer,

Thank you very much for your careful review and valuable suggestions. Our manuscript has been checked by a native English-speaking and corrected for language and style. In addition, the following changes have been made to the manuscript. We highlighted in blue all the changes in the main text.

Comment 1:

The abstract section should be improved. The authors should follow the manuscript's structure: Background and aims, Material and methods, results, discussion, and conclusion. Moreover, it should be a total of about 200 words maximum. Please, change it accordingly, following the authors' guidelines (https://www.mdpi.com/journal/antioxidants/instructions).

Response 1:

We have improved the abstract section in accordance with the guidelines and have kept the word count at about 200 words.

Comment 2:

The introduction section is too long: please, summarize the main background information in order to introduce the theme, describing clearly the aims. Please, revisit it.

Response 2:

We have shortened the introduction section appropriately to introduce the theme and describe the aims more clearly.

Comment 3:

Despite the material and methods section describe the main tools utilized in this study, it should be improved. I suggest inserting a schematic picture in order to clarify the experimental model. Moreover, the authors missed inserting several important pieces of information (the composition of each group, the middle age, the health status, the diet conditions, ....). Please, improve this section.

Response 3:

In order to clarify the experimental model, we have inserted a schematic picture in line 456. We have added the composition, sex, age, weight, health status, and diet conditions of experimental animals in lines 87-92.

Comment 4:

The results section should be improved. In Figure 2, the metric reference should be inserted (bar line). The statistical analysis should be completely described in supplementary files. Please, insert complete information in order to better analyze the discussed results.

Response 4:

In Figure 2, we've inserted clear bar lines. In order to better analyze, complete descriptions for statistical analysis have also been inserted in lines 184-191. Please notify us in detail if the changes do not meet the requirements.

Comment 5:

Finally, the original images (inserted as a supplementary file) of the blots are strange: please, insert the gel images at the moment of the acquisition, without increasing the contrast of the bands.

Response 5:

The WB images in the supplementary file are the original images at the moment of acquisition without contrast adjusted.

Comment 6:

The discussion section should be improved: the authors should insert several considerations about the limitations of the present study.

Response 6:

We have improved the discussion section by adding some considerations about the limitations of this study in lines 450-454.

Round 2

Reviewer 2 Report

The authors substantially didn't address my primary concern. They should show their images with a good magnification both in figure 2 and figure 4. Besides, they didn't indicate the magnification measures for the photos in 4A. Therefore, they should improve at least the presented data if they cannot offer new ones.

Author Response

Dear reviewer,

Thank you for your reply again. We have adjusted the magnification of Figure 2 and Figure 4A for better viewing. We have also added the scale bar for Figure 4A and inserted the magnification of Figure 4A on line 263. We hope the changes meet your comments.

Reviewer 3 Report

The authors have made almost all requested changes. Nevertheless, I have doubts about the images (inserted as a supplementary file) of the blots: please, insert the gel images at the moment of the acquisition, without increasing the contrast of the bands. The part inserted in the main text is the same as the full picture. Please, solve this important criticism.

Author Response

Dear reviewer,

Thank you for your reply again. We have replaced all WB images in the text with the same ones as the original. We hope the changes meet your comments.
